# Mechanosensitive Channel PIEZO1 Senses Shear Force to Induce KLF2/4 Expression via CaMKII/MEKK3/ERK5 Axis in Endothelial Cells

**DOI:** 10.3390/cells11142191

**Published:** 2022-07-13

**Authors:** Qi Zheng, Yonggang Zou, Peng Teng, Zhenghua Chen, Yuefeng Wu, Xiaoyi Dai, Xiya Li, Zonghao Hu, Shengjun Wu, Yanhua Xu, Weiguo Zou, Hai Song, Liang Ma

**Affiliations:** 1Department of Cardiovascular Surgery, The First Affiliated Hospital, School of Medicine, Zhejiang University, Hangzhou 310009, China; 3130101980@zju.edu.cn (Q.Z.); 1516066@zju.edu.cn (P.T.); 22118165@zju.edu.cn (Z.C.); dxy1995@zju.edu.cn (X.D.); wsjsw@zju.edu.cn (S.W.); 2The MOE Key Laboratory of Biosystems Homeostasis & Protection, Zhejiang Provincial Key Laboratory for Cancer Molecular Cell Biology and Innovation Center for Cell Signaling Network, Life Sciences Institute, Zhejiang University, Hangzhou, Zhejiang 310058, China; zyg951226@zju.edu.cn (Y.Z.); flaviuswu@zju.edu.cn (Y.W.); 12107102@zju.edu.cn (X.L.); 11707042@zju.edu.cn (Z.H.); 0919276@zju.edu.cn (Y.X.); 3CAS Center for Excellence in Molecular Cell Sciences, State Key Laboratory of Cell Biology, Shanghai Institute of Biochemistry and Cell Biology, University of Chinese Academy of Sciences, Shanghai 200031, China

**Keywords:** PIEZO1, KLF2/4, endothelial cell, mechanosensitive transcription factor, shear stress, signal transduction

## Abstract

Shear stress exerted by the blood stream modulates endothelial functions through altering gene expression. KLF2 and KLF4, the mechanosensitive transcription factors, are promoted by laminar flow to maintain endothelial homeostasis. However, how the expression of KLF2/4 is regulated by shear stress is poorly understood. Here, we showed that the activation of PIEZO1 upregulates the expression of KLF2/4 in endothelial cells. Mice with endothelial-specific deletion of *Piezo1* exhibit reduced KLF2/4 expression in thoracic aorta and pulmonary vascular endothelial cells. Mechanistically, shear stress activates PIEZO1, which results in a calcium influx and subsequently activation of CaMKII. CaMKII interacts with and activates MEKK3 to promote MEKK3/MEK5/ERK5 signaling and ultimately induce the transcription of *KLF2/4*. Our data provide the molecular insight into how endothelial cells sense and convert mechanical stimuli into a biological response to promote KLF2/4 expression for the maintenance of endothelial function and homeostasis.

## 1. Introduction

Endothelial cells lining the inner layer of blood vessels regulate vascular functions and maintain vascular homeostasis. Multiple factors, such as hormones, cytokines, neurotransmitters and physical forces created by the blood flow, determine the complexities of endothelial structure and function [1,2,3,4]. Different blood flow patterns, including unidirectional laminar flow (UF) with high laminar shear stress in straight vessels and disturbed flow (DF) at curved regions, bifurcations and branch points, modulate distinct endothelial functions and morphology. Endothelial cells respond to the changes of local shear stress to modulate the intracellular signaling network, which leads to the alterations of gene expression and cell morphology [5,6,7]. It has been well documented that abnormal shear stress, predominantly DF, leads to endothelial inflammation and dysfunction, eventually causing various cardiovascular diseases, including atherosclerosis, aortic dissection and hypertension [8,9,10,11], which are the leading cause of mortality worldwide [12].

Emerging evidence shows that distinct flow patterns play different roles in the function of endothelial cells mainly through mechanosensitive transcription factors (MSTFs), among which the Kruppel-like factor (KLF2) and the Kruppel-like factor 4 (KLF4) are the best-characterized and act as the master regulators of endothelial-related genes [13,14]. KLF2/4, promoted by UF but downregulated by DF, provide vascular protective effects to maintain endothelial homeostasis through regulating the expression of multiple anti-inflammatory genes, such as eNOS and THBD [15,16]. In addition, UF-induced KLF2 recruits the transcriptional co-activator p300 to inhibit the activation of NF-κB, which is the leading proinflammatory MSTF upregulated by DF [17]. However, how the expression of KLF2 and KLF4 is regulated by different flow patterns remains elusive.

Mechanosensitive ion channels sense mechanical force in the cell membrane and rapidly convert it into electrical or chemical signals. As a critical mechanosensitive ion channel in endothelial cells, PIEZO1 plays diverse roles in endothelial shear stress sensing, nitric oxide (NO) generation, vascular tone, angiogenesis, atherosclerosis, vascular permeability and remodeling, blood pressure regulation, etc [18,19,20,21,22,23,24]. It has been reported that depending on the flow pattern, PIEZO1 transduces mechanical signals to exert different cellular processes as either an atheroprotective or as an inflammatory signal. Both PIEZO1 and KLF2/4 respond to different patterns of blood flow. However, the possible relationship between them and the underlying molecular mechanism remain unclear. Here, we showed that the activation of endothelial PIEZO1 regulates the expression of KLF2 and KLF4 through a Ca^2+^/CaMKII/MEKK3/MER5/ERK5 axis. PIEZO1-mediated Ca^2+^ influx is the initial factor for the mechanotransduction process by laminar flow, resulting in the activation of downstream kinase and ultimately inducing *KLF2/4* transcriptional expression, thus suppressing the NF-κB signaling pathway to provide an anti-inflammatory effect and maintain endothelial homeostasis.

## 2. Materials and Methods

### 2.1. Cell Culture and Treatment

HEK293T cells and bEnd.3, immortalized mouse brain microvascular endothelial cells (MBMECs), were cultured in DMEM (Invitrogen) supplemented with 10% fetal bovine serum (FBS; Gibco) and 50 μg/mL penicillin/streptomycin. Human umbilical vein endothelial cells (HUVECs) were purchased from Sciencell and cultured in Endothelial Cell Medium (ECM; Sciencell) containing 5% FBS and 50 μg/mL penicillin/streptomycin. All cells were cultured under a condition of 5% CO_2_ at 37 °C.

MBMECs were seeded with a density of 40,000 cells per cm^2^. After serum starvation overnight, MBMECs were treated with agonists, including Yoda1 (TargetMol), 2-APB (100 μM; TargetMol), GSK1016790A (10 μM; TargetMol) and ICILIN (10 μM; TargetMol) or pretreated with inhibitors, including KN-93 (10 μM, 2 h, TargetMol), TAE226 (10 μM, 2 h, TargetMol), Bisindolylmaleimide I (10 μM, 2 h, TargetMol), BIX02189 (10 μM, 12 h, TargetMol), BAPTA-AM (10 μM, 2 h, TargetMol) and Ruthenium Red (RR, 10 μM, 12 h, TargetMol). For the time-course experiments, the concentration of Yoda1 was 5 μM, and for all dose-gradient experiments, the treated time of Yoda1 was 8 h for Western blot and 2 h for quantitative RT-PCR unless otherwise stated.

### 2.2. Transfection and Infection

HEK293T cells at 70% confluence were transfected with control plasmid or plasmids containing Flag-MEKK3 and HA-CaMKII by PEI (Polyscience). siRNA transfection was performed by LipoRNAiMAX (Thermo Fisher, Rockford, IL, USA). *Piezo1*-KO MBMECs and *Mekk3*-KO MBMECs were generated using Lenti-CRISPR-V2 vector (Addgene #52961). siRNA and sgRNA sequences are listed in Appendix A.

### 2.3. Shear Stress Application

Shear stress was applied to MBMECs by an orbital shaker (Thermo Fisher) inside the incub ator as previously described [25]. MBMECs were seeded in a single well of a 6-well plate with a diameter of 3.5 cm and cultured with 2 mL medium. After cell attachment, the plate was cultured on a shaker at 210 rpm for 5 days. Disturbed flow was applied to MBMECs in the center with a diameter of 12 mm, and laminar flow was applied in the periphery within 8 mm from the edge. Control cells were exposed to static conditions. Microfluidic Circulatory System with a peristaltic pump was built according to the published method [26]. Microfluidic system mimics the fluid shear stress of bloodstream by controlling the flow rate. MBMECs were seeded in a flow chamber (ibidi 80176) to form monolayers. After cell attachment, the flow chamber was connected to the flow system, and laminar flow was applied with 16 dyn/cm^2^. Poiseuille’s equation was used to calculate the shear stress that cells experienced τ = 4Qη/πR^3^.

### 2.4. Ca^2+^ Imaging

MBMECs were loaded with Fluo-4 AM (2 μM, Beyotime, Shanghai, China) for 1 h at 37 °C in DMEM and then washed out by PBS. Yoda1 (5 μM) was added into medium, and cells were incubated at static conditions for 10 min. Image was acquired by a Nikon NI-U fluorescent microscope.

### 2.5. Immunofluorescence Staining

MBMECs were seeded on round cover slide and transfected with control or plasmids containing Flag-MEKK3 or HA-CaMKII by lipofectamine 3000 (Thermo Fisher) for 6 h. After 24 h culture, cells were treated with Yoda1 (5 μM) or DMSO and then fixed in 4% PFA for 20 min followed by permeabilization (0.1% Triton X-100 in PBS) for 10 min and blocking in 5% bovine serum albumin (BSA) in PBS for 1 h at room temperature. Cells were then incubated with primary antibody against Flag (1:100; Sigma, Burlington, MA, USA, F3165) and HA (1:100; Proteintech, 51064-2-AP) for 2 h at room temperature. After being washed with PBS, cells were then incubated with corresponding secondary antibodies (1:1000; Alexa Fluor 488- or 594-conjugated antibodies; Jackson ImmunoResearch Laboratories, West Grove, PA, USA) for 1 h at room temperature followed by DAPI (0.5 μg/mL) staining for detection of nuclei at room temperature for 15 min in the dark. Images were taken by a confocal microscope (LSM880).

For cytoskeleton staining, 4% PFA fixed cells were incubated with phalloidin (1:100; Actin-Tracker Green; Beyotime, Shanghai, China) for 1 h. Images were taken by a Nikon NI-U fluorescence microscope.

### 2.6. Western Blotting and Co-Immunoprecipitation

After treatment, cells were collected in RIPA buffer containing protease and phosphatase inhibitor cocktail on ice. Equal amounts of proteins were loaded in SDS-PAGE and then transferred to PVDF membranes (EMD Millipore). After blocking in 5% dried skimmed milk or BSA, the membranes were incubated with primary antibodies at 4 °C overnight with gentle agitation. Then, membranes were incubated with horseradish peroxidase (HRP)-conjugated secondary antibodies for 1 h at room temperature. Proteins were visualized using an enhanced chemiluminescent substrate (ECL) and a chemiluminescence imaging system (ChemiScope5600, Clinx, Shanghai, China). Rabbit anti-phosph-MEKK3 antibody was made by immunized rabbit with peptide: CSGTGMR(P)SVTGTPYW. Antibody information is described in Appendix A.

For co-immunoprecipitation assay, cells were lysed with lysis buffer (50 mM Tris-HCl, pH 7.4, 100 mM NaCl, 10% glycerol, 0.5% NP-40, 1 mM DTT, 1 mM PMSF). Anti-HA magnetic beads (Thermo Fisher, 88837) were added to the cell lysates and incubated at 4 °C overnight with gentle agitation. Beads were washed by lysis buffer three times, and proteins were eluted by SDS loading buffer for western blotting.

### 2.7. Reverse Transcription (RT) and Quantitative RT-PCR Analysis

Total RNA was extracted using RNAiso plus reagent (Takara), followed by quantification with NanoDrop 2000 Spectrophotometer (Thermo Fisher). Equal RNA was reverse-transcribed with PrimeScript^TM^ RT Master Mix (Takara), and then real-time PCR was perfumed using SYBR Green (Vazyme, Nanjing, China) on a fluorescence quantitative PCR system (CFX96 Connect, Bio-Rad). Expression levels were analyzed using CT values and compared to control group. The qPCR primer sequences are shown in Appendix A.

### 2.8. Animals

The *Piezo1*^fl/fl^ mice were described previously [27]. Endothelial-specific *Tie2Cre* (Tg(Tek-cre)1Ywa/J, Strain #:008863) mice were from the Jackson Laboratory. *Piezo1*^fl/fl^ mice and *Tie2Cre* mice were used to generate compound mutants. Male ICR mice with a body weight of approximate 25 g were purchased from Shanghai SLAC Laboratory Animal Company. Animals were housed in SPF facility under a 12 h light/dark cycle at Zhejiang University Laboratory Animal Center. All animal experiments were approved by the Institutional Animal Care and Use Committee of Zhejiang University.

### 2.9. Isolation of Mouse Aortas and Brain Endothelial Cells and Drug Treatment

GdCl_3_ (20 mg/kg, Shanghai, China) or KN-93 (2 mg/kg, Shanghai, China) was given into wild-type male ICR mice by tail vein injection every 12 h for 4 times, and thoracic aortas were collected 12 h after the last injection. Mice were sacrificed via cervical dislocation. Thoracic aortas were isolated as previously described [28] and followed by RNA isolation. Briefly, mice were sacrificed via cervical dislocation, and chest cavity was opened with dissection scissors, and abdominal aorta was cut to release the blood. Thoracic aorta was removed using micro-dissection forceps and was flushed gently with ice-cold PBS to remove the blood. The attached adipose tissue and small lateral vessels were removed as much as possible using micro-dissection forceps.

Mouse primary brain endothelial cells were isolated by collagenase digestion and density gradient centrifugation using low molecular weight dextran. Briefly, mouse brain was dissected into 2 mm pieces and homogenized with a homogenizer. To the homogenate, a similar volume of 30% dextran was added to give a 15% dextran solution, followed by centrifugation at 3000× *g* for 15 min at 4 °C. The pellet was resuspended and digested in 0.1% collagenase/dispase and 0.1% DNase I in DMEM medium for 45 min at 37 °C. The cell pellet was resuspended in EBM-2 medium (Lonza CC-3156) containing 5% serum and transferred to culture dishes pretreated with 1% gelatin.

### 2.10. Histology and Immunofluorescence

Mouse lungs were isolated and fixed with 4% PFA at 4 °C for 6 h, followed by dehydration with ethanol and were cleared with xylene and embed in paraffin. Sections were cut at 6 μm thickness and used for immunofluorescence with primary antibody overnight at 4 °C, followed by biotinylated secondary antibody and HRP-conjugated streptavidin (1:1000; Jackson ImmunoResearch Laboratories) in combination with fluorogenic substrate Alexa Fluor 488 tyramide (1:300; TSA kit, PerkinElmer, Montgomeryville, PA, USA). Sections were heated in microwave for 3 min and then incubated with another primary antibody CD31 (1:200; Abcam, Cambridge, UK ab182981) at 4 °C overnight and followed by Alexa Fluor 594-conjugated secondary antibody (1:1000; Jackson ImmunoResearch Laboratories) at room temperature for 2 h in the dark. Nucleus was stained with DPAI (0.5 μg/mL) at room temperature for 15 min. Images were taken by a confocal microscope (LSM880).

### 2.11. RNA Sequencing and Data Analysis

Total RNA extracted from MBMECs was quantified and qualified with NanoDrop 2000 Spectrophotometer (Thermo Fisher). The cDNA libraries were sequenced using BGIseq500 platform (BGI, Shenzhen, China). The raw sequence data were trimmed and checked with the FastQC and Trimmatic. The mouse genome sequence file and mouse genome annotation file were downloaded from the ensemble official server. The hisat2 was used to align the reads back to the reference genome by exon. Then, we managed to convert and sort SAM generated by hisat2 file to BAM file through samtools. Finally, the htseq outputted the counts. The differential-expressed genes (DEGs) were identified with DEseq2 under selecting criteria of adjusted *p* value less than 0.05 and absolute value of fold change larger than 2 (negative binomial generalized linear model fitting and Wald test). 

The counts were normalized by converting them into Fragments Per Kilobase Million (FKPM). The corresponding log10 FKPM values for DEGs were used for heat map plotting with clusters between different Yoda1-treated time groups. Gene Ontology (GO) and Kyoto Encyclopedia of Genes and Genomes (KEGG) pathway enrichment were performed with Clusterprofiler. In the enrichment result, *p* < 0.05 or FDR < 0.05 is a meaningful pathway. The protein interactions for differential-expressed gene overlaps from three comparisons were obtained from STRING database. The hub genes were calculated by MCC in cytoscape. If not otherwise stated, all the analyses were performed with R 4.1.2.

### 2.12. Statistical Analysis

All data are shown as the mean ± SEM (Standard Error of Mean). Statistical analysis was performed using unpaired Student’s *t* test by GraphPad Prism version 8. Unpaired Welch’s *t*-test was applied for the comparisons of means between two groups, and one-way ANOVA test and Tukey post hoc test were performed for the comparisons among multiple groups. Statistical significance was assumed if *p* values were less than 0.05. 

## 3. Results

### 3.1. Shear Stress and Activation of PIEZO1 Enhance KLF2/4 Expression in Endothelial Cells

Previous studies have shown that UF plays a critical role in endothelial homeostasis by inducing the expression of KLF2/4 [29,30]. To investigate the regulatory mechanism of shear stress on the expression of KLF2/4, we performed an in vitro shear stress experiment by culturing immortalized mouse brain microvascular endothelial cells (MBMECs) in microfluidic chambers applied with ~16 dyn/cm^2^ shear stress for 3 and 6 h. We found that the expression of *Klf2* mRNA was rapidly induced at 3 h after UF stimulation and followed by *Klf4* induction at 6 h (Figure 1A). In addition, shear stress was achieved by a rotating orbital shaker at 210 rpm in which cells were exposed to ~5 dyn/cm^2^ shear stress in the center of the well to simulate DF and ~12 dyn/cm^2^ shear stress on the periphery to stimulate UF (Appendix A). The morphology of MBMECs considerably changed under different flow patterns compared with ST (static) culture after 5 days (Appendix A). Consistent with a previous study, the mRNA and protein levels of KLF2 and KLF4 were markedly increased in MBMECs after exposure to UF-induced high-shear stress compared with DF conditions (Figure 1B,C). The phosphorylation level of YAP was also increased under UF (Appendix A) as previously reported [31,32,33]. In addition, we found a shear force-dependent KLF4 expression under UF by increasing the rotating speed (Appendix A). KLF2/4 are highly expressed in the endothelial cells at the thoracic aorta [34], an area exposed to a high-speed blood flow, resembling a high-shear stress condition. To test whether mechanically activated ion channels sense shear stress to induce *Klf2* and *Klf4* expression in vivo, we injected GdCl_3_, a non-specific mechanically activated ion channel blocker, into mice to inhibit these channels. To reduce the renal toxicity of GdCl_3_ in the mice [35], we collected whole mouse thoracic aortas after four doses of GdCl_3_ injection within 2 days. We found that *Klf2* and *Klf4* mRNA levels were decreased significantly after GdCl_3_ treatment (Figure 1D). Although GdCl_3_ is likely to have other effects not limited to mechanical channel blockade, this result suggests that there is a correlation between the blockade of mechanically activated channels and reduced *Klf2/4* mRNA expression. Taken together, our data suggest that UF-induced *Klf2*/*4* expression in endothelial cells is mediated by mechanically activated ion channels.

To sort out which mechanically activated ion channels regulate the shear stress-induced *Klf2*/*4* expression, we used several mechanosensitive ion channel agonists, including Yoda1 (PIEZO1 agonist), 2-APB (TRPV1/2/3 agonist), GSK1016790A (TRPV4 agonist) and ICILIN (TRPM8 agonist), to activate these channels in MBMECs. Of note, only Yoda1 treatment activated the expression of KLF4 in both a time- and dose-dependent manner (Figure 1E–L). Consistently, Yoda1-induced KLF4 was also observed in primary human umbilical vein endothelial cells (HUVECs) (Appendix A). To further verify PIEZO1-mediated regulation of endothelial *Klf2*/*4* in response to shear stress, we cultured the freshly isolated thoracic aortas in vitro for 24 h to eliminate the effects of shear stress on the endothelial cells of the aortas. We found that the mRNA levels of *Klf2* were markedly reduced after static culture of the thoracic aortas (Figure 1M). Although *Klf4* mRNA levels remained unchanged during the culture, both *Klf2* and *Klf4* mRNA levels were increased when the thoracic aortas were treated with Yoda1 in the culture medium (Figure 1M). Together, these data indicate that the activation of PIEZO1 induces KLF2/4 expression in endothelial cells. 

### 3.2. Deletion of Piezo1 in Endothelial Cells Suppresses Shear Stress-Mediated KLF2/4 Expression 

To further investigate whether *Piezo1* is required for shear stress-induced endothelial *Klf2*/*4* expression in vivo, we generated endothelial cell-specific *Piezo1* knockout *Tie2Cre*;*Piezo1*^fl/fl^ (EC-KO) mice. *Tie2Cre* activity is reported to be pan-endothelial cell by E9.5 and remains such throughout development [36]. *Tie2Cre* mouse models show some degree of Cre recombinase activity in non-endothelial cell expression, such as the hematopoietic lineage and heart valves [37]. EC-KO mice were born at a Mendelian ratio and appeared normal. Because *Piezo1* is highly expressed in the endothelium, we collected thoracic aortas from EC-KO mice at P10 to examine the expression of *Piezo1* mRNA. Quantitative RT-PCR showed that the *Piezo1* gene was effectively removed in the endothelial cells of thoracic aortas (Figure 2A). Importantly, the mRNA levels of *Klf2* and *Klf4* were significantly decreased in the thoracic aortas of EC-KO mice (Figure 2A). Consistently, a decreased KFL4 protein expression was also observed in the endothelial cells of lung blood vessels from EC-KO mice by immunofluorescence staining (Figure 2B). Of note, the expression of KLF4 in non-vascular tissue of the lungs remained unchanged (Figure 2B). In addition, we isolated brain endothelial cells from WT and *Piezo1* EC-KO mice and examined the expression of *Klf2/4.* Consistently, the mRNA levels of *Klf2* and *Klf4* were also significantly decreased in mouse brain endothelial cells (Figure 2C). Next, we created pooled *Piezo1*-depleted MBMECs by lentivirus-mediated CRISPR/Cas9 targeting *Piezo1* (Figure 2D). Consistently, UF-induced *Klf2* and *Klf4* expression was greatly reduced in *Piezo1*-deficient MBMECs (Figure 2E,F). Moreover, Yoda1-induced KLF4 expression was significantly inhibited by the depletion of *Piezo1* in MBMECs (Figure 2G and Appendix A), which confirmed the specificity of Yoda1 on targeting PIEZO1. In addition, the time- and dose-dependent *Klf2*/*4* induction by Yoda1 were reduced in *Piezo1*-deficient MBMECs (Figure 2H and Appendix A, respectively). Together, these data suggest that endothelial PIEZO1 acts as a specific mechanosensitive ion channel to mediate KLF2/4 expression in response to shear stress both in vitro and in vivo. 

### 3.3. Mekk3 Deletion in Endothelial Cells Restrains PIEZO1-Mediated KLF2/4 Expression

KLF2 and KLF4 have been demonstrated to be regulated by the MEKK3/MEK5/ERK5 signaling pathway in the developing mouse heart and vascular endothelium [38,39]. We asked whether the MEKK3 (also known as MAP3K3) signaling pathway is involved in the regulation of KLF2/4 expression in endothelial cells by shear stress. To test this, we first performed an in vitro shear stress experiment. After exposed to shear stress, increased phosphorylation of MEKK3 and EKR5 was readily observed in MBMECs (Figure 3A). Notably, UF induced more phosphorylation of MEKK3 and EKR5 than DF (Figure 3A). To examine whether PIEZO1 could promote the activation of MEKK3/MEK5/ERK5 signaling, we treated MBMECs with Yoda1 to activate PIEZO1. The results showed that the phosphorylation of MEKK3 and ERK5 was increased after Yoda1 treatment in a dose-dependent manner (Figure 3B). These data suggest that the activation of PIEZO1 stimulates the activation of the MEKK3/ERK5 signaling pathway.

To further investigate whether MEKK3 is required for the PIEZO1-mediated KLF2/4 expression, we generated *Mekk3*-knockout (*Mekk3*-KO) MBMECs by CRISPR/Cas9. The induction of KLF4 protein under shear stress was strongly inhibited after loss of *Mekk3* expression in MBMECs (Figure 3C). A similar result was observed at the mRNA level by quantitative analysis of *Klf2* and *Klf4* mRNA expression (Figure 3D). Furthermore, Yoda1-induced *Klf2* and *Klf4* expression was also inhibited in *Mekk3*-KO MBMECs (Figure 3E,F and Appendix A). To address the possible role of MEK5/ERK5 activity in the regulation of *Klf2*/4 expression, we pretreated MBMECs with a specific MEK5 inhibitor (BIX02189) before Yoda1 administration. Consistent with the observation in *Mekk3*-KO MBMECs, Yoda1-induced *Klf2* and *Klf4* expression was strongly reduced when MEK5 activity was inhibited (Figure 3G,H and Appendix A). Of note, the basal levels of *Klf2* and *Klf4* were decreased in *Mekk3*-KO and MEK5 inhibitor-treated MBMECs (Figure 3F,H), which highlights a central role of MEKK3/MEK5/ERK5 signaling in regulating *Klf2*/*4* expression in endothelial cells. Thus, these results suggest that PIEZO1-induced KLF2/4 expression is dependent on the MEKK3/MEK5/ERK5 signaling pathway.

### 3.4. Inhibition of CaMKII Suppresses PIEZO1-Mediated KLF2/4 Expression

As a mechanosensitive Ca^2+^ permeable ion channel in endothelial cells, PIEZO1 participates in the sensing of shear stress and responds to mechanical stimuli through regulating the Ca^2+^ influx [18,40]. To unravel the molecular mechanism, which regulates shear stress-induced *Klf2*/*4* expression, we explored whether Ca^2+^ influx is involved in this process. To test this, we first examined the Ca^2+^ influx caused by PIEZO1 activation after Yoda1 treatment using the Ca^2+^ sensitive dye Fluo-4 AM. Activation of PIEZO1 by Yoda1 resulted in a rapid Ca^2+^ influx in control MBMECs, but not in *Piezo1*-deficient MBMECs (Figure 4A). To test whether the Ca^2+^ influx is necessary for the induction of *Klf2*/*4* expression during Yoda1 treatment, we pretreated MBMECs with a selective calcium chelator, BAPTA-AM (10 μM), for 2 h before Yoda1 administration. As expected, the induction of KLF4 was drastically inhibited by the BAPTA-AM pretreatment (Figure 4B Appendix A). A similar result was observed with the pretreatment of an L-type Ca^2+^ current blocker, Ruthenium Red (RR). Yoda1-induced KLF4 was absent in MBMECs pretreated with RR (Figure 4C and Appendix A). To further confirm the effect of Ca^2+^ influx in endothelial cells under shear stress exposure, we cultured MBMECs in an orbital shaker with or without RR treatment. Consistently, the shear stress-induced *Klf2* and *Klf4* expression was significantly reduced after RR treatment (Figure 4D). These data suggest that calcium is a critical secondary messenger to mediate PIEZO1-induced KLF2/4 expression in endothelial cells.

Calcium activates calcium/calmodulin-dependent protein kinases (CaMKs), protein kinases C (PKCs) and focal adhesion kinases (FAKs), which have been shown to play an important role in shear stress-mediated endothelial stability [41,42,43]. Next, we tested whether these kinases act as downstream of PIEZO1-induced Ca^2+^ influx to regulate *Klf2*/*4* expression. MBMECs were pretreated with various calcium-dependent protein kinase inhibitors to block the corresponding kinases. Both KN-93 (a CaMKII inhibitor) and TAE226 (a FAK inhibitor) pretreatment diminished the expression of KLF4 induced by Yoda1, but not the PKC inhibitor (Figure 4E). Since CaMKII has been shown to be regulated by PIEZO1 in macrophages [44], we then focused on the role of CaMKII in PIEZO1-induced *Klf2/4* expression. MBMECs were exposed to different types of shear stress. We found that only UF induced the phosphorylation of CaMKII, indicating the activation of CaMKII (Figure 4F). Consistently, the pretreatment with KN-93, a CaMKII-specific inhibitor, reduced the Yoda1-mediated induction of KLF2 and KLF4 in both MBMECs and primary HUVECs (Figure 4G and Appendix A). Interestingly, actin depolymerization was also observed in MBMECs after KN-93 treatment (Appendix A). To further examine the role of CaMKII in the regulation of endothelial *Klf2* and *Klf4* expression in vivo, wild-type ICR mice were injected with KN-93 through the tail vein, and thoracic aortas were collected for *Klf2* and *Klf4* mRNA quantification. We found that KN-93 treatment significantly reduced the mRNA levels of *Klf2*/*4* in mouse thoracic aorta endothelial cells (Figure 4H). Together, these data indicate that CaMKII is indispensable to transduce PIEZO1-induced Ca^2+^ influx to activate KLF2/4 expression.

### 3.5. CaMKII Interacts with and Activates MEKK3

Our results revealed that both the PIEZO1/Ca^2+^/CaMKII and PIEZO1/MEKK3/MEK5/ERK5 axes transcriptionally regulated the expression of *Klf2*/*4*, which prompted us to ask whether there is a link between CaMKII and MEKK3. Interestingly, KN-93 administration diminished the ERK5 activation induced by Yoda1 treatment in MBMECs (Figure 5A,B), which suggests that CaMKII might be a key component upstream of MEKK3 to transduce PIEZO1-mediated signals. To further investigate whether CaMKII directly regulates MEKK3, we performed a co-immunoprecipitation assay in HEK293T cells using tagged MEKK3 combined with various CaMKII isoforms. The results reveal that MEKK3 interacted with all isoforms of CaMKII, among which CaMKIIγ had the most potent interaction (Figure 5C). An immunofluorescence assay revealed the co-localization of tagged MEKK3 and CaMKIIγ in the cytoplasm of MBMECs (Figure 5D). Furthermore, Proximity Ligation Assay (PLA) was performed to investigate the interaction between MEKK3 and CaMKIIγ in MBMECs. Strong PLA fluorescence signals were only observed throughout the cytoplasm of MBMECs transfected with both FLAG-MEKK3 and HA-CaMKIIγ plasmids, but not in control cells (Figure 5E). To examine whether PIEZO1 activation regulates the interaction between CaMKII and MEKK3, we treated these MBMECs with Yoda1 for 1 and 2 h. We found that Yoda1 treatment promoted the interaction of CaMKIIγ with MEKK3 as indicated by the increased PLA signal over time (Figure 5F). Importantly, the activation of MEKK3 indicated by the increased phosphorylation of MEKK3 was only observed in HEK293T cells expressing kinase-active mutant CaMKIIγ-T287E, but not in cells expressing kinase-dead mutant CaMKIIγ-T287A (Figure 5G). Taken together, these results suggest that CaMKII interacts with MEKK3 and activates MEKK3 kinase activity.

### 3.6. PIEZO1 Restrains the TNF-Induced NF-κB Activation in Endothelial Cells through KLF2/4

To comprehensively assess the transcriptional regulation mediated by PIEZO1 activation in endothelial cells, we performed an RNA-sequencing analysis using Yoda1-stimulated MBMECs. Consistently, a volcano plot analysis showed that the mRNA levels of *Klf2* and *Klf4* significantly increased after 2 h of Yoda1 treatment (Figure 6A). A Venn diagram analysis revealed that *Klf2* and *Klf4* were the common genes that were increased at each time point during Yoda1 treatment (Figure 6B). Interestingly, a Maximal Clique Centrality (MCC) analysis revealed that KLF2 and KLF4 are the hubs of the complex interactome from the commonly induced genes during the time-course Yoda1 treatment (Figure 6B), highlighting the central role of KLF2 and KLF4 in PIEZO1-mediated biological processes in endothelial cells. Notably, a KEGG pathway analysis revealed that the TNF and NF-κB signaling pathways were activated in the initial period, but downregulated after prolonged Yoda1 treatment (Figure 6C and Appendix A). As mechanosensitive transcription factors, KLF2 and KLF4 also exert an anti-inflammatory role in endothelial cells by inhibiting NF-κB transcriptional activity [45]. Thus, these observations prompted us to investigate whether UF-mediated PIEZO1 activation provides an anti-inflammatory effect by promoting the expression of KLF2 and KLF4. To test this, we treated MBMECs with TNF to activate NF-κB signaling, in the presence of Yoda1, which mimics PIEZO1 activation by laminar flow. TNF-induced p65 phosphorylation was compromised of Yoda1 pretreatment (Figure 6D). Immunoblot confirmed that KLF4 was effectively activated by Yoda1 during TNF treatment (Figure 6D). Consistently, the induction of *A20* mRNA, a classic downstream of the NF-κB signaling pathway, by TNF treatment was strongly reduced by Yoda1 pretreatment (Figure 6E). Importantly, the knockdown of *Klf2* and *Klf4* in MBMECs using siRNA prevented the inhibitory effect on NF-κB activation caused by Yoda1 pretreatment in MBMECs, indicated by the rebounded phosphorylation of p65 and the expression of *A20* (Figure 6F,G). These results suggest that PIEZO1 activated by Yoda1 plays an anti-inflammatory role in endothelial inflammation by stimulating KLF2/4 expression.

## 4. Discussion

It was well recognized that mechanical signals dynamically control gene transcription programs to determine cell fate and organogenesis during development [46,47]. However, it is still less understood how the signaling cascade links the membrane mechanosensitive channel to the nuclear transcription machinery. Endothelial cells respond to shear stress to exert biological functions and pathological process through various signaling pathways and MSTFs [8,48]. KLF2/4, upregulated by UF but suppressed by DF, have been demonstrated to be the most vital MSTFs in endothelial cells, which preserve endothelial homeostasis by regulating the expression of multiple anti-inflammatory and anti-oxidant genes [15,16,49]. In this study, we investigated the mechanism of how KLF2/4 are regulated by the shear force. We found that PIEZO1 activates the expression of *Klf2/4* in endothelial cells in response to laminar shear stress. Our results show that the PIEZO1/Ca^2+^/CaMKII/MEKK3/MEK5/ERK5 signaling cascade promotes the expression of endothelial *Klf2*/*4* responding to laminar flow by the pharmaceutical approach and animal model (Figure 7). Our results described here demonstrate how mechanical cues modulate the activities of nuclear factors that dictate transcription programs to control cell behaviors and the fate in the endothelium.

Aberrant mechanical signaling has been associated with the pathogenesis of multiple diseases. The development and homeostasis of the vascular system are intimately connected to mechanical forces associated with blood flow and blood pressure. Shear stress in aortas exerts diametrically opposed effects in a flow pattern-dependent manner. UF in straight thoracic and abdominal aortas maintains endothelial homeostasis by preserving endothelial cell structure, alignment and quiescence, which therefore provides anti-inflammatory effects. However, DF in the aortic arch and atherosclerotic plaque regions leads to endothelial dysfunction and impairs vascular permeability, which thus exerts pro-inflammatory effects [6,8,50,51]. In our study, high laminar shear stress and a higher concentration of Yoda1 promoted the expression of *Klf2/4*. To this end, the molecular mechanism of how different patterns of blood flow promote distinct PIEZO1 functions remains unclear. Perhaps this is related to the conformational changes of PIEZO1 when it senses different forms of blood flow. An alternative mechanism might be that PIEZO1 can be enhanced by other factors or post-transcriptional modification, such as phosphorylation.

One study reported *Piezo1* deficiency relieved atherosclerosis in LDL receptor-deficient (LDLR^−/−^) mice [21], which is contrary to its anti-inflammatory effect under UF from our cell-based study. Previous studies have revealed that KLF2 and KLF4 have an anti-inflammation function, which thereby protects aortic vessels from atherosclerosis [15,52]. However, besides KLF2/4, hemodynamic forces activate several signaling pathways to regulate endothelial phenotypes associated with atherosclerosis, via other transcription factors, such as YAP/TAZ, NRF2, HIF-1a, NF-κB and AP-1 [48]. Perhaps the pro-inflammatory effect of PIEZO1 under DF in the atherosclerosis animal model is caused by the dysfunction of those transcription factors. Indeed, we found that the activation of PIEZO1 by Yoda1 suppressed YAP activity by promoting its phosphorylation. In addition, Yoda1 treatment initially enhanced NF-κB activity, but long-time treatment resulted in the recession of NF-κB by inducing KLF2/4 expression. It is very likely that these transcription factors crosstalk with one another to regulate endothelial inflammation upon exposure to hemodynamic forces. Understanding how KLF2/4 work in concert with these MSTFs in endothelial homeostasis will be beneficial for us to gain a panoramic view of the mechanical sensing signaling network in endothelial cells.

Endothelial calcium influx is induced by PIEZO1 under shear stress, which maintains endothelial focal adhesion and alignment [18,53]. Consistent with a role of Ca^2+^ influx in preservation of endothelial homeostasis, CaMKII is the key regulator in endothelial functions, including NO induction, vascular permeability, endothelial migration and proliferation [54,55,56,57]. A recent study also showed that CaMKII was upregulated by PIEZO1 to adjust endothelial permeability [58]. Here, we found that endothelial CaMKII, activated by shear stress and PIEZO1, promoted MEKK3 activation and KLF2/4 expression. Therefore, we established a novel crosstalk between the CaMKII and MAPK signaling pathways. We have recently shown that MEKK3 activity is restricted by the STRIPAK complex [59]. The STRIPAK complex is a multicomponent supramolecule, which is composed of phosphatases, kinases and other components, among which the striatin family is the scaffold of the complex. The striatin family has been shown to bind Ca^2+^-calmodulin (CaM) in the presence of Ca^2+^ through its CaM-binding domain [60,61]. In addition, the STRIPAK complex also associates with MEKK3 through CCM2 and CCM3 [59]. Endothelial loss of CCM2/3 results in the gain of MEKK3-KLF2/4 signaling and causes the cerebral cavernous malformation disease. Whether the STRIPAK complex is regulated by Ca^2+^ and is involved in the activation of CaMKII/MEKK3 axis and the expression of *Klf2/4* is worth further investigation.

Endothelial PIEZO1 is involved in multiple physiologies and various pathologies. PIEZO1 acts as the molecular pressure sensors to regulate blood pressure, and deletion of both PIEZO1 and PIEZO2 in mice leads to the complete absence of barosensation and baroreceptor reflex function [62]. Another key role of PIEZO1 in the endothelium is the regulation of nitric oxide (NO) release, which is an important molecular to adjust vasodilation and endothelial homeostasis [20].

One of the cardiovascular diseases involved in endothelial PIEZO1 is pulmonary arterial hypertension (PAH), which is characterized by an increase in pressure and mechanical forces inside the intrapulmonary arteries [63]. Piezo1 mediates arterial relaxation by promoting the production of NO in PAH and use of GsMTx-4-ameliorated experimental pulmonary hypertension in vivo [64]. However, little is known about the fully pathogenic role of PIEZO1 in pulmonary arterial hypertension. Further experiments are required to demonstrate the potential role of Yoda1 in the chronic hypoxia-induced pulmonary hypertension mouse model.

Considering the pro-inflammatory effect of PIEZO1 under DF, two studies have shown that Piezo1 deficiency relieved atherosclerosis in LDL receptor-deficient (*LDLR^−/−^*) mice [21], and pharmacological inhibition of Piezo1 by GsMTx-4 administration attenuated plaque formation in *ApoE^−/−^* mice [65]. However, there is currently less reported about the located application with appropriate concentration of Yoda1 in the atherosclerosis model, given its anti-inflammatory effect in UF. Our research revealed the specific relationship between PIEZO1 and KLF2/4 in the UF-mediated anti-inflammatory process. Activation of PIEZO1 by Yoda1 that mimics laminar shear stress inhibited the phosphorylation of P65 and the expression of downstream inflammatory factors induced by TNF. This anti-inflammatory effect of Yoda1 suggests its potential pre-clinical and clinical use in atherosclerosis. However, further studies are still needed to prove the clinical implications and translational potential of the mechanism that PIEZO1 regulates KLF2/4 expression in endothelial functions and cardiovascular diseases by the *LDLR^−/−^* and *ApoE^−/−^*-induced atherosclerosis mice model. 

Although our study identifies the mechanism that mechanical cues modulate KLF2/4 expression through PIEZO1, we did not test the function of this mechanism in disease models, such as atherosclerosis, hypertension and thoracic aortic aneurysm, or in specimens from patients with cardiovascular disease, which is a critical step for clinical translation. Moreover, the CRISPR/Cas9 system in primary endothelial cells is not yet possible due to poor efficiency and cytotoxicity, which led us to perform experiments using immortalized endothelial cell lines or treating primary HUVECs with various inhibitors to validate the results.

## 5. Conclusions

Emerging evidence connects MSTF dysregulation by mechanical cues to various human vascular diseases. We have discovered a novel endothelial mechanical signaling pathway consisting of PIEZO1 and a kinase cascade CaMKII/MEKK3/MEK5 that links fluid shear stress to the nuclear transcription factors KLF2 and KLF4. Although the underlying mechanisms remain to be further defined in many cases, understanding the MSTF dysregulation by mechanical cues in human diseases potentially can provide insights into new therapeutic strategies for these diseases.

## Figures and Tables

**Figure 1 cells-11-02191-f001:**
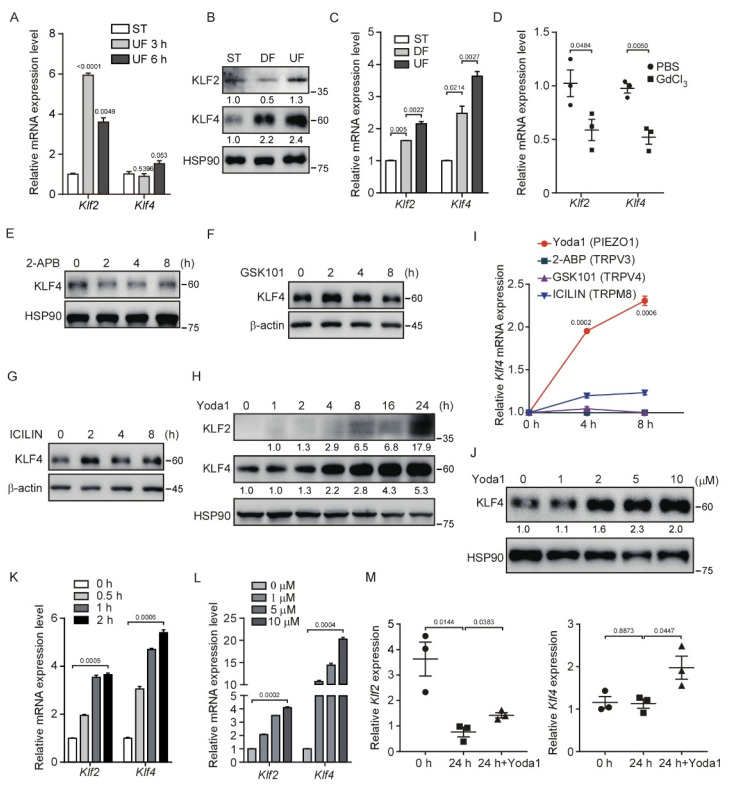
Shear stress induces KLF2/4 expression through PIEZO1 activation. (**A**) MBMECs were seeded in a flow chamber and exposed to unidirectional laminar flow with 16 dyn/cm^2^ for indicated time, and mRNA levels of *Klf2* and *Klf4* were measured by quantitative RT-PCR. (**B**,**C**) MBMECs were exposed to differential shear stress for 5 days. KLF2 and KLF4 proteins were monitored by western blotting (**B**), and mRNA levels of *Klf2* and *Klf4* were measured by quantitative RT-PCR (**C**). (**D**) Wild-type male ICR mice were injected with GdCl_3_ (20 mg/kg) via tail vein every 12 h for 4 times, and thoracic aortas were collected for quantitative RT-PCR. (**E**–**H**) MBMECs were treated with indicated mechanosensitive ion channel agonists including 2−APB (TRPV1/2/3 agonist, 100 μM), GSK1016790A (TRPV4 agonist, 10 μM), ICILIN (TRPM8 agonist, 10 μM) and Yoda1 (PIEZO1 agonist, 5 μM) for the indicated time. Protein (**E**–**H**) levels of KLF4 were monitored by western blotting, respectively, and quantified with ImageJ (**I**). (**J**) MBMECs were treated with Yoda1 with the indicated concentration for 8 h, and KLF4 proteins were monitored by western blotting. (**K**) MBMECs were treated with Yoda1 (5 μM) for the indicated time, and mRNA levels of *Klf2* and *Klf4* were measured by quantitative RT-PCR. (**L**) MBMECs were treated with Yoda1 with the indicated concentration for 2 h, and mRNA levels of *Klf2* and *Klf4* were measured by quantitative RT-PCR. (**M**) The thoracic aortas from wild-type male ICR mice were harvested and cultured in different conditions. mRNA levels of *Klf2* and *Klf4* in thoracic aortas were measured by quantitative RT-PCR. At 0 h, total RNA was extracted immediately after aortas were isolated. At 24 h, total RNA was extracted after aortas were cultured in the culture medium in vitro without shear stress exposure for 24 h. At 24 h + Yoda1, aortas were cultured with Yoda1 (5 μM) treatment in culture medium for 24 h in vitro. *n* = 3 mice/group. Data are representative of three independent experiments and are presented as mean ± SEM of three technical replicates by an unpaired Student’s *t*-test (**A**,**C**,**I**,**K**,**L**). The immunoblot was measured using ImageJ to determine the relative intensities of the indicated bands, which were normalized using the internal loading control proteins. The relative intensities are shown as numbers. ST: static culture; DF: disturbed flow; UF: unidirectional laminar flow.

**Figure 2 cells-11-02191-f002:**
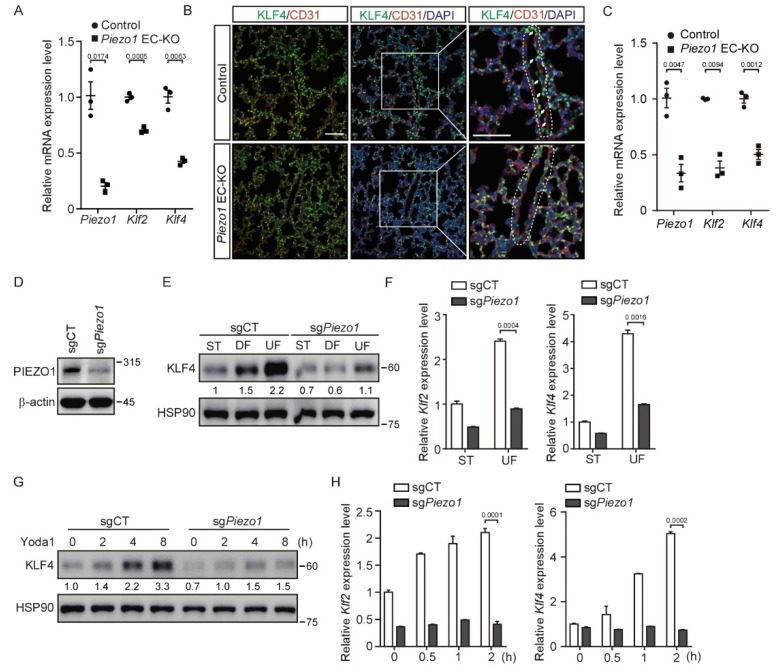
Deletion of PIEZO1 in endothelial cells impairs KLF2/4 expression induced by shear stress. (**A**) Thoracic aortas of *Piezo1*^fl/fl^ and *Tie2Cre*;*Piezo1*^fl/fl^ mice at P10 were collected, and mRNA expression levels of *Piezo1*, *Klf2* and *Klf4* were measured by quantitative RT-PCR. n = 3 mice/each group. (**B**) Representative immunofluorescence staining of lungs from *Piezo1*^fl/fl^ and *Tie2Cre*;*Piezo1*^fl/fl^ mice in the same litter for CD31 (red), KLF4 (green) and nuclei (blue). Arrows indicate the KLF4 expressing endothelial cells. Dotted lines outline the blood vessel. Scale bars are 50 μm. n = 3 mice/each group. (**C**) Primary mouse brain endothelial cells from *Piezo1*^fl/fl^ and *Tie2Cre*;*Piezo1*^fl/fl^ mice at 4 weeks were isolated, and mRNA levels of *Piezo1*, *Klf2* and *Klf4* were measured by quantitative RT-PCR. n = 3 mice/each group. (**D**) Control and *Piezo1*-dificient MBMECs were harvested, and cell lysates were probed with anti-PIEZO1 antibody. (**E**,**F**) Control and *Piezo1*-dificient MBMECs were exposed to shear stress for 5 days. KLF4 proteins (**E**) and mRNA levels of *Klf2* and *Klf4* (**F**) were monitored by western blotting and quantitative RT-PCR, respectively. (**G**,**H**) Control and *Piezo1*-dificient MBMECs were treated with Yoda1 (5 μM) for the indicated time. KLF4 proteins (**G**) and mRNA levels of *Klf2* and *Klf4* (**H**) were monitored by western blotting and quantitative RT-PCR, respectively. Data are representative of three independent experiments and are presented as mean ± SEM of three technical replicates by an unpaired Student’s *t*-test (**F**,**H**). The immunoblot was measured using ImageJ to determine the relative intensities of the indicated bands, which were normalized using the internal loading control proteins. The relative intensities are shown as numbers. ST: static culture; UF: unidirectional laminar flow.

**Figure 3 cells-11-02191-f003:**
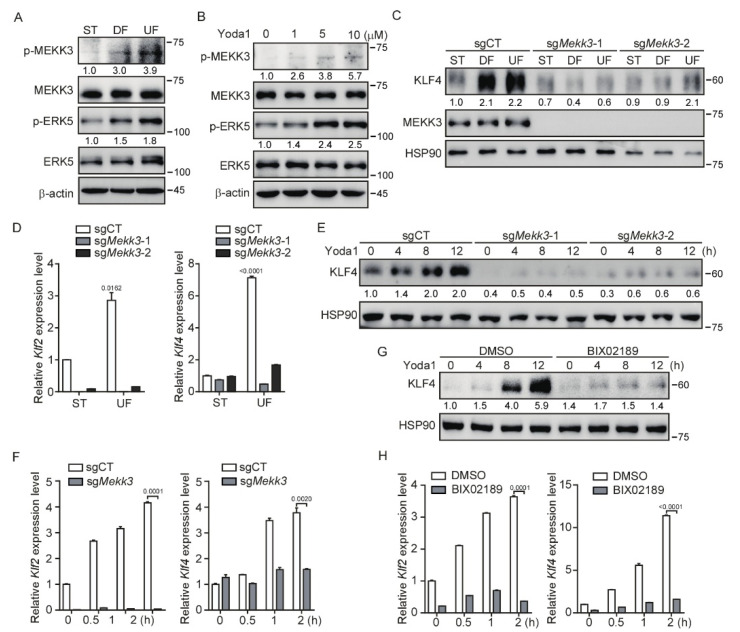
*MEKK3* deletion in endothelial cells restrains PIEZO1-induced KLF2/4 expression. (**A**) MBMECs were exposed to differential shear stress for 5 days, and cell lysates were probed with the indicated antibodies. (**B**) MBMECs were treated with Yoda1 with the indicated concentration for 2 h, and cell lysates were probed with the indicated antibodies. (**C**,**D**) Control and *Mekk3*-dificient MBMECs were exposed to differential shear stress for 5 days. KLF4 and MEKK3 proteins were monitored by western blotting (**C**), and mRNA levels of *Klf2* and *Klf4* were measured by quantitative RT-PCR (**D**). (**E**,**F**) Control and *Mekk3*-dificient MBMECs were treated with Yoda1 (5 μM) for the indicated time, and KLF4 proteins (**E**) and mRNA levels of *Klf2* and *Klf4* (**F**) were monitored by western blotting and quantitative RT-PCR respectively. (**G**,**H**) MBMECs were pretreated with MEK5 inhibitor (BIX02189, 10 μM) for 12 h and then treated with Yoda1 (5 μM) for the indicated time. KLF4 proteins (**G**) and mRNA levels of *Klf2* and *Klf4* (**H**) were monitored by western blotting and quantitative RT-PCR, respectively. Data are representative of three independent experiments and are presented as mean ± SEM of three technical replicates by an unpaired Student’s *t*-test (**D**,**F**,**H**). The immunoblot was measured using ImageJ to determine the relative intensities of the indicated bands, which were normalized using the internal loading control proteins. The relative intensities are shown as numbers. ST: static culture; UF: unidirectional laminar flow.

**Figure 4 cells-11-02191-f004:**
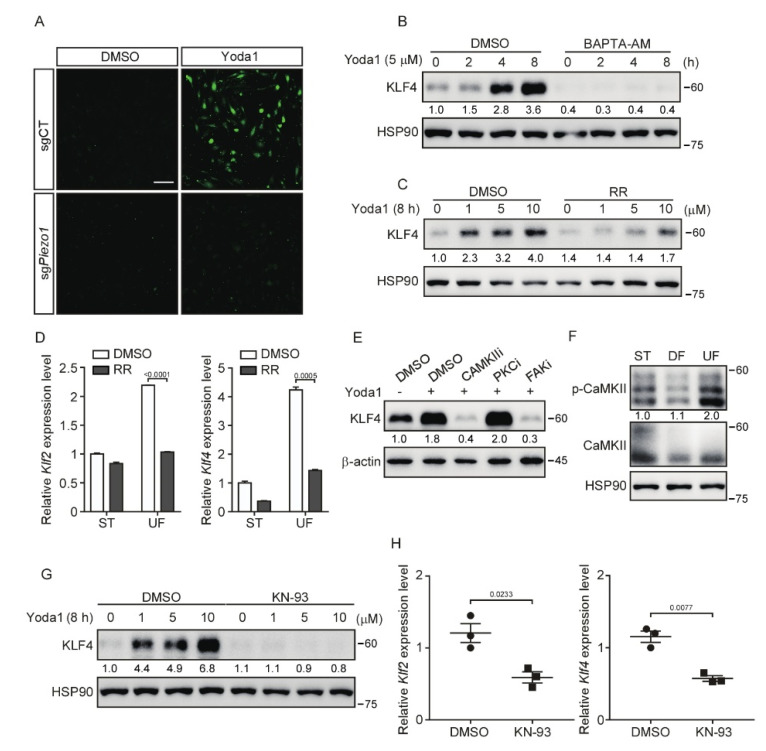
Inhibition of CaMKII suppresses PIEZO1/MEKK3-mediated KLF2/4 expression. (**A**) Control and *Piezo1*-dificient MBMECs were pretreated with Fluo-4 AM (2 μM) for 1 h and then treated with Yoda1 (5 μM) for 10 min. Calcium imaging was analyzed under a fluorescence microscope. Scale bars, 50 μm. (**B**) MBMECs were pretreated with calcium chelator (BAPTA-AM, 10 μM) for 2 h and then treated with Yoda1 (5 μM) for the indicated time. KLF4 proteins were monitored by western blotting. (**C**) MBMECs were pretreated with L-type calcium current blocker, Ruthenium Red (RR, 10 μM), for 2 h and then treated with Yoda1 with the indicated concentration for 8 h. KLF4 proteins were monitored by western blotting. (**D**) MBMECs were treated with RR (10 μM) in presence of shear stress exposure for 5 days, and mRNA levels of *Klf2* and *Klf4* were measured by quantitative RT-PCR. (**E**) MBMECs were pretreated with CaMKII inhibitor (KN-93, 10 μM), PKC inhibitor (Bisindolylmaleimide I, 10 μM) and FAK inhibitor (TAE226, 10 μM) for 2 h and then treated with Yoda1 (5 μM) for 8 h. KLF4 proteins were monitored by western blotting. (**F**) MBMECs were exposed to differential shear stress for 5 days, and cell lysates were probed with indicated antibodies. (**G**) MBMECs were pretreated with KN-93 (10 μM) for 2 h and then treated with Yoda1 (5 μM) for 8 h. KLF4 proteins were monitored by western blotting. (**H**) Wild-type male ICR mice were injected with KN-93 (2 mg/kg) via tail vein every 12 h for 4 times, and thoracic aortas were collected for quantitative RT-PCR of *Klf2* and *Klf4* mRNA expression. n = 3 mice/group. Data are representative of three independent experiments and are presented as mean ± SEM of three technical replicates by an unpaired Student’s *t*-test (**D**). The immunoblot was measured using ImageJ to determine the relative intensities of the indicated bands, which were normalized using the internal loading control proteins. The relative intensities are shown as numbers. ST: static culture; UF: unidirectional laminar flow.

**Figure 5 cells-11-02191-f005:**
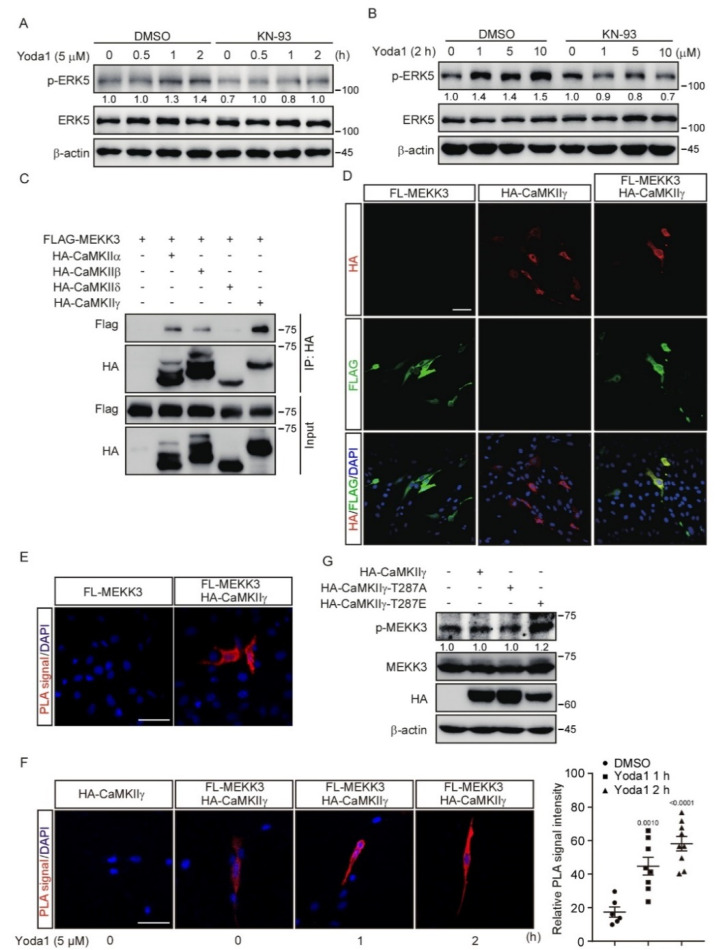
MEKK3 signaling pathway is activated via PIEZO1-mediated CaMKII activation. (**A**,**B**) MBMECs were treated with KN−93 (10 μM, 2 h) and Yoda1 as indicated. Cell lysates were probed with the indicated antibodies. (**C**) HEK293T cells were transfected with the indicated plasmids and lysed for co-immunoprecipitation assay using an anti−HA antibody, followed by western blotting using the indicated antibodies. (**D**) Representative immunofluorescence staining with anti−HA antibody (red) and anti−FLAG (green) antibody in MBMECs transfected with the indicated plasmids. Scale bars, 50 μm. (**E**,**F**) MBMECs were transfected with the indicated plasmids. Cells were treated with Yoda1 (5 μM) or DMSO (**F**) for the indicated times 48 h after transfection. PLA was used to detect the interaction between FLAG−MEKK3 and HA−CaMKIIγ in MBMECs. Relative PLA signal intensity of cells in each group was calculated by ImageJ and shown in right panel of F. PLA signal: red, DAPI: blue. Scale bars, 50 μm. (**G**) HEK293T cells were transfected with the indicated plasmids, and cell lysates were probed with the indicated antibodies. The immunoblot was measured using ImageJ to determine the relative intensities of the indicated bands, which were normalized using the internal loading control proteins. The relative intensities are shown as numbers.

**Figure 6 cells-11-02191-f006:**
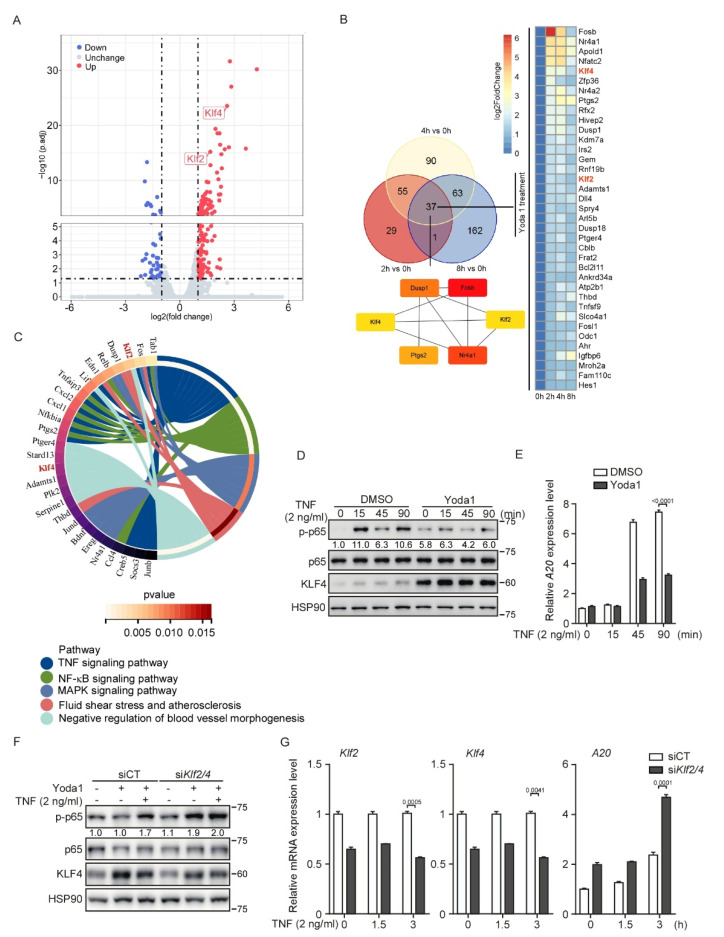
PIEZO1 restrains the TNF-induced NF−κb activation in endothelial cells. (**A**) The volcano plot for transcriptome changes of MBMECs after 2 h Yoda1 treatment (5 μM) was built by defining *x*-axis as the fold change values and *y*-axis as P-adjust. Red dots indicate upregulated genes; blue dots indicate downregulated genes; grey dots indicate not significant change. (**B**) Venn diagram shows the overlap genes significantly changed after Yoda1 (5 μM) treatment for 2, 4 or 8 h, respectively, in MBMECs. Maximal Clique Centrality (MCC) analysis was at the lower panel of Venn diagram. The lines indicate the interactions. The color shows the rank calculated by MCC. More essential genes have deeper red color. (**C**) Circle enrichment shows that the genes are linked via ribbons to their assigned signaling pathways or biological functions after 2 h Yoda1 treatment in MBMECs. White-to-red coding on right hand side indicates the *p* value of selected terms. Color next to the selected genes indicates number of pathways to which the genes contribute. (**D**,**E**) MBMECs were pretreated with Yoda1 (5 μM) for 8 h and then treated with TNF (2 ng/mL) for the indicated time. Cell lysates were probed with the indicated antibodies (**D**), and mRNA levels of *A20* were measured by quantitative RT−PCR (**E**). (**F**,**G**) Control and siRNA-mediated *Klf2/4* knockdown MBMECs were pretreated with Yoda1 (5 μM) for 4 h and then treated with TNF (2 ng/mL) for 3 h. Cell lysates were probed with the indicated antibodies (**F**), and mRNA levels of *Klf2, Klf4* and *A20* were measured by quantitative RT−PCR (**G**). Data are representative of three independent experiments and are presented as mean ± SEM of three technical replicates by an unpaired Student’s *t*-test (**E**,**G**). The immunoblot was measured using ImageJ to determine the relative intensities of the indicated bands, which were normalized using the internal loading control proteins. The relative intensities are shown as numbers.

**Figure 7 cells-11-02191-f007:**
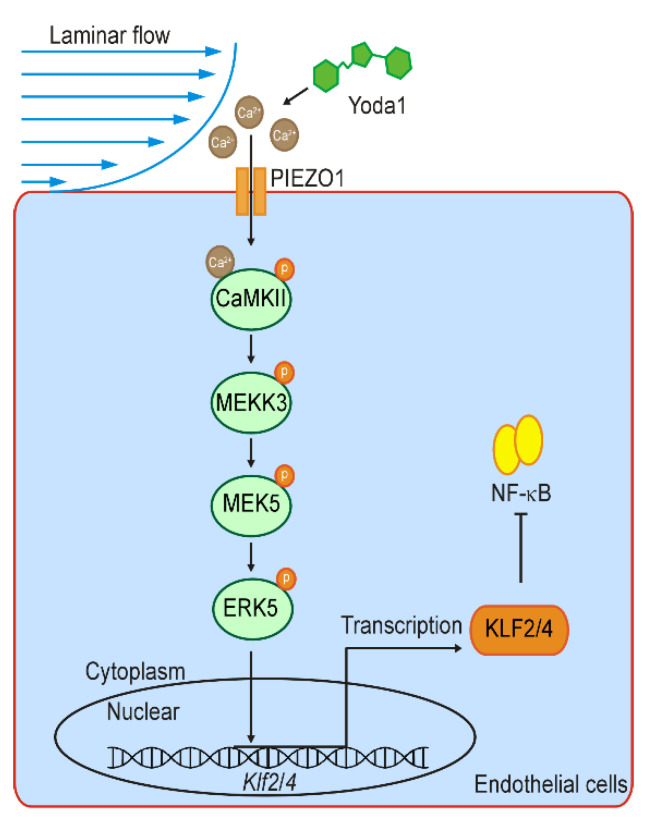
Schematic diagram shows the shear stress-induced KLF2/4 expression via PIEZO1/CaMKII/MEKK3/ERK5 cascade. Laminar flow stimulates the activation of PIEZO1 and thus causes Ca^2+^ influx, which further activates CaMKII. Cascade phosphorylation of MEKK3/MEK5/ERK5 is induced by the activation of CaMKII and ultimately leads to *Klf2* and *Klf4* transcription. Elevated KLF2/4 inhibits NK-κB to exert anti-inflammatory effect in the endothelium response to laminar flow.

## Data Availability

The datasets are available from the corresponding author on reasonable request. All data needed to evaluate the conclusions in the paper are present in the paper. The raw RNA-Seq data accompanied with this paper have been uploaded to the Genome Sequence Archive in the National Genomics Data Center, China National Center for Bioinformation (CNCB)/Beijing Institute of Genomics (BGI), Chinese Academy of Sciences (CAS) and are publicly accessible at https://ngdc.cncb.ac.cn/gsa under the access number CRA006427.

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
