# Peer review of "Mechanosensitive Channel PIEZO1 Senses Shear Force to Induce KLF2/4 Expression via CaMKII/MEKK3/ERK5 Axis in Endothelial Cells"

_cells, 2022, doi:10.3390/cells11142191_

Round 1

Reviewer 1 Report

The authors have performed a study showing that flow (shear stress) activates the Klf2/4 in endothelial cells through a pathway involving the activation by Piezo of the CaMKII/MEKK3/ERK5 cascade. The authors used both in vitro and in vivo models to show this pathway. The study is done properly, and the data presented is convincing. Nevertheless, some concerns remain, as described below.

Major:

1 – Use of the orbital shaker: This system produces a flow which changes progressively from the periphery to the center. The use of flow chambers allows to produce a true laminar flow and a well-defined disturbed flow (oscillatory flow being the most used). The authors should reproduce some of their key results using a flow chamber with a fully laminar flow compared to an oscillatory flow generated in well-controlled flow chamber.

2 - Tie2Cre mice were used to target the endothelium. Nevertheless, bone marrow cells also express Tie2 and the authors should also discuss this as a limitation of the study.

3 – Statistics: non-parametric tests should be used with small N numbers. 

3 – GdCl3 was injected in vivo and this is likely to have numerous effects not limited to channels blockade. Even with the only blockade of channels, many pathways are affected. Thus, claiming that the effect of Gd on klf2/4 expression is mediated by mechanically activated ion channels is difficult to accept. Did the authors check for renal toxicity (urea, creatinine for example)?  Also add a sentence in the discussion to mention this limitation.

Minor:

1 – Why did the authors use UF to describe laminar flow. LF would be easier to understand.

2 – The authors should discuss the limitation of the use of MBMECs which are very different from native endothelial cells. 

3 - Define ST, UF, DF in the figure legends to make them self-explanatory.

Author Response

On behalf of all the contributing authors, I sincerely thank the editor and the reviewers for their enthusiasm regarding our study and the insightful comments to improve the quality of this study. We performed additional experiments to address all the points raised by the reviewers. We have extensively revised our manuscript accordingly and marked revised manuscript with the changes. We hope that the changes we have made would resolve the reviewer’s concerns about the manuscript.

The authors have performed a study showing that flow (shear stress) activates the Klf2/4 in endothelial cells through a pathway involving the activation by Piezo of the CaMKII/MEKK3/ERK5 cascade. The authors used both in vitro and in vivo models to show this pathway. The study is done properly, and the data presented is convincing. Nevertheless, some concerns remain, as described below.

 Major:

1 – Use of the orbital shaker: This system produces a flow which changes progressively from the periphery to the center. The use of flow chambers allows to produce a true laminar flow and a well-defined disturbed flow (oscillatory flow being the most used). The authors should reproduce some of their key results using a flow chamber with a fully laminar flow compared to an oscillatory flow generated in well-controlled flow chamber.

Response: We appreciate the reviewer’s comments. The same issue was raised by other 2 reviewers. As reviewer’s suggestion, we have added the experiment using microfluidic system to mimics the fluid shear stress of bloodstream by controlling the flow rate in the revised Figure 1A. We found that the expression of Klf2 mRNA was rapidly induced at 3 hours after UF stimulation and followed by Klf4 induction at 6 hours with ~16 dyn/cm2 shear stress Figure 1A. This result was consistent with the data using rotating orbital shaker. We also added the experimental procedure in the revised Materials and Methods.

2 - Tie2Cre mice were used to target the endothelium. Nevertheless, bone marrow cells also express Tie2 and the authors should also discuss this as a limitation of the study.

Response: We appreciate the reviewer’s comments on the limitation of Tie2Cre model. As reviewer’s suggestion, we have added the information about the Tie2Cre in the revised manuscript at page 6 as the following sentences. Tie2Cre activity is reported to be pan-endothelial cell by E9.5 and remains such throughout development [36]. Tie2Cre mouse models show some degree of Cre recom-binase activity in Non-endothelial cell expression, such as the hematopoietic lineage and heart valves [37].

3 – Statistics: non-parametric tests should be used with small N numbers. 

Response: Thank you for your concern on the sample size. The t-test can be used for small sample size comparison (n<30). The three biological replicates are commonly seen in a pile of publications. The non-parametric test may be more robust. However, as the general audience of this paper is not the bioinformatics expert. The over-explanation of the statistic methods will make paper redundant and cause the unnecessary misunderstanding. We have added detailed statistical analysis method in the revised manuscript.

4 – GdCl3 was injected in vivo and this is likely to have numerous effects not limited to channels blockade. Even with the only blockade of channels, many pathways are affected. Thus, claiming that the effect of Gd on klf2/4 expression is mediated by mechanically activated ion channels is difficult to accept. Did the authors check for renal toxicity (urea, creatinine for example)?  Also add a sentence in the discussion to mention this limitation.

Response: We appreciate the reviewer’s concern about the nonspecific effect of GdCl3. We agree with the reviewer’s comment. GdCl3 is widely used in the research of mechanically activated ion channels. Our GdCl3 study established a link of mechanically activated ion channels with KLF4 expression. Our cell-based study revealed that Piezo1 is required for shear force-induced KLF4 expression. Further, we used Tie2Cre; Piezo1-KO mice to show that Piezo1 regulates KLF4 expression in endothelial cells in vivo. We have discussed the limitation of GdCl3 in animal study, such as non-specific effects and renal toxicity the revised manuscript at page 6 as the following sentences. To reduce the renal toxicity of GdCl3 to the mice [35], we collected whole mouse thoracic aortas after 4 doses of GdCl3 injection for 2 days. Although GdCl3 is likely to have other effects not limited to mechanical channels blockade, this result suggests that there is a correlation between the blockade of mechanically activated channels and reduced Klf2/4 mRNA expression.

Minor:

1 – Why did the authors use UF to describe laminar flow. LF would be easier to understand.

Response: Both UF and LF are used in the literature. We have spelled out the abbreviations for UF and DF in the manuscript and figure legends to make it easier for readers to understand.

2 – The authors should discuss the limitation of the use of MBMECs which are very different from native endothelial cells. 

Response: We agree with the reviewer’s comment. We have added the discussion of the limitation of the use of MBMEC cells at page 17 as the following sentences. Although our study identifies the mechanism that mechanical cues modulate KLF2/4 expression through PIEZO1, we did not test the function of this mechanism in disease models, like atherosclerosis, hypertension and thoracic aortic aneurysm, or in specimens from patients with cardiovascular disease, which is a critical step for clinical translation. Moreover, CRISPR/Cas9 system in primary endothelial cells are not yet possible due to poor efficiency and cytotoxicity, which led us to perform experiments using immortalized endothelial cell lines or treating primary HUVECs with various inhibitors to validate the results.

3 - Define ST, UF, DF in the figure legends to make them self-explanatory.

Response: We apologize for the inconsistency. We have corrected it.

Reviewer 2 Report

This is an eloquent and thorough examination by Zheng, Zou, and Teng et al. that details Piezo-1 dependent signaling in endothelial cells. The authors were able to clearly show a strong dependence of the CAMKII/MEKK3/ERK5 signaling axis on Piezo-1 in vitro and establish that KLF2/4 are major endpoint regulators with the study culminating in RNA sequencing analyses that led to the identification of Piezo-1 in an anti-inflammatory role, also mediated by KLFs. I have only minor requests and comments. Please see below:

Figure S1D. Reporting the calculated shear instead of or in addition to the rotating speed is preferred.

Immunoblots should be quantified using densitometry and compared using group statistics throughout the study. Similarly, quantification of the PLA with statistical comparisons would be beneficial.

It is assumed that the sgCT and siCT controls used in this study are scrambled but this is not explicitly clear. The sequences of the controls should also be listed in the Supplementary Table 1.

Figure 5. Does UF (or Yoda-1) promote an increase in CAMKII/MEKK3 interactions? It is interesting and novel that these two cellular players interact, but the role of Piezo-1 is unclear here.

Figure 6. The authors state that activation of Piezo-1 via Yoda-1 or laminar shear play an anti-inflammatory role, however, only Yoda-1 was used in these experiments. Please rephrase as the effects of laminar shear can only be inferred here.

Figure 6F. It appears as though KLF2/4 knockdown results in an elevated p-p65 independent of TNF-mediated activation of NF-Kb which would perhaps indicate baseline regulation of the inflammatory signaling. Quantification of the blots would be beneficial to clarify.

Author Response

On behalf of all the contributing authors, I sincerely thank the editor and the reviewers for their enthusiasm regarding our study and the insightful comments to improve the quality of this study. We performed additional experiments to address all the points raised by the reviewers. We have extensively revised our manuscript accordingly and marked revised manuscript with the changes. We hope that the changes we have made would resolve the reviewer’s concerns about the manuscript.

This is an eloquent and thorough examination by Zheng, Zou, and Teng et al. that details Piezo-1 dependent signaling in endothelial cells. The authors were able to clearly show a strong dependence of the CAMKII/MEKK3/ERK5 signaling axis on Piezo-1 in vitro and establish that KLF2/4 are major endpoint regulators with the study culminating in RNA sequencing analyses that led to the identification of Piezo-1 in an anti-inflammatory role, also mediated by KLFs. I have only minor requests and comments. Please see below:

Figure S1D. Reporting the calculated shear instead of or in addition to the rotating speed is preferred.

Response: We appreciate the reviewer’s comments. The same issue was raised by other 2 reviewers. As reviewer’s suggestion, we have added the experiment using microfluidic system to mimics the fluid shear stress of bloodstream by controlling the flow rate in the revised Figure 1A. We found that the expression of Klf2 mRNA was rapidly induced at 3 hours after UF stimulation and followed by Klf4 induction at 6 hours with ~16 dyn/cm2 shear stress Figure 1A. This result was consistent with the data using rotating orbital shaker. We also added the experimental procedure in the revised Materials and Methods.

Immunoblots should be quantified using densitometry and compared using group statistics throughout the study. Similarly, quantification of the PLA with statistical comparisons would be beneficial.

Response: As reviewer’s suggestion, we have added the quantification of immunoblots and PLA signals in the revised figures.

It is assumed that the sgCT and siCT controls used in this study are scrambled but this is not explicitly clear. The sequences of the controls should also be listed in the Supplementary Table 1.

Response: As reviewer’s suggestion, we have added the control sequences in the revised Supplementary Table 1.

Figure 5. Does UF (or Yoda-1) promote an increase in CAMKII/MEKK3 interactions? It is interesting and novel that these two cellular players interact, but the role of Piezo-1 is unclear here.

Response: As reviewer’s suggestion, we performed additional experiment to examine whether Piezo1 regulates the interaction of CAMKII and MEKK3 in the revised Figure 5F. We treated MBMEC cells overexpressed tagged CAMKII and MEKK3 with Yoda1 to activate PIEZO1 and examine the interaction between CAMKII and MEKK3 by proximity ligation assay (PLA). We found that the PLA signal was significantly increased after Yoda1 treatment, which indicates that PIEZO1 activation promotes the interaction between CAMKII and MEKK3.

Figure 6. The authors state that activation of Piezo-1 via Yoda-1 or laminar shear play an anti-inflammatory role, however, only Yoda-1 was used in these experiments. Please rephrase as the effects of laminar shear can only be inferred here.

Response: We apologize for the misleading statement. We have rewritten the sentence at Page 9.

Figure 6F. It appears as though KLF2/4 knockdown results in an elevated p-p65 independent of TNF-mediated activation of NF-Kb which would perhaps indicate baseline regulation of the inflammatory signaling. Quantification of the blots would be beneficial to clarify.

Response: We appreciate for the reviewer’s comment. Several studies have demonstrated that KLF2/4 inhibit NF-κB activity. In our study, we also observed the increased baseline activity of NF-κB pathway. As reviewer’s suggestion, we have quantified the immunoblots in the revised manuscript.

Reviewer 3 Report

Here, Zheng and co-authors investigated the influence of mechanosensitive PIEZO1 channel on the regulation of atheroprotective transcription factors KLF2 and KLF4.  Overall, in vitro and in vivo experiments have been nicely performed and WBs are well complemented with qPCR data. The article is methodologically sound and is logically written. The authors employed an impressive armamentarium, including genetically engineered Cre-Lox mice with the endothelial-specific Piezo1 knockout and RNA sequencing of Yoda1-stimulated microvascular endothelial cells. All images and gels are of excellent quality.

The authors performed an important and timely discovery, convincingly showing the indispensable role of Piezo1 mechanosensitive channel in controlling the expression of KLF2 and KLF4, the major endothelial atheroprotective factors. This is particularly important when considering KLF2/4 agonists as promising tools to prevent the development of atherosclerosis at the gateway (i.e., to prevent endothelial dysfunction). PIEZO1 is a major mechanosensitive channel which can be finely targeted through the specific agonists (e.g., Yoda1 as was shown in the paper). The results can be therefore applicable in pre-clinical studies on ApoE or LDLR -/- mice.

My only significant concern is the statistics, as authors performed an incorrect parametric analysis (yet this does not deny the study results). The authors should revise the statistical analysis according to the respective comment (#2).

The paper can be accepted for publication upon the minor revision and is strongly recommended to be highlighted on the journal site or cover page.

Below are the specific comments:

1. Rotational shaker is not the best option to induce shear stress as it is barely possible to calculate the exact shear stress values (dyn/cm2) in this experimental setting. A much better option is the cell culture under unidirectional flow perfectly imitating the blood flow (e.g. ibidi Pump System Quad), as it permits to set the exact shear stress values and even to simulate oscillatory and laminar flow. Yet, the authors ensured the disturbed or laminar flow by means of assessing the cell geometry.

2. The use of parametric statistics is incorrect for being used here, as the number of replicates is too low. Use non-parametric statistics instead (median, interquartile range, Kruskal-Wallis test, Mann-Whitney U-test). In any case, the use of standard error instead of standard deviation is also incorrect as it does not reflect the sample variation. Please also describe (and perform if needed) the adjustment for multiple comparisons (Tukey's post hoc test? False discovery rate?) Please also indicate all P values in a numerical manner, not as asterisks.

3. Please describe how you isolated RNA from mouse aortic endothelium to confirm the loss of PIEZO1 expression (Figure 2A).

4. Please comment on the clinical implications and translational potential of the study in the Discussion, as the study surely provides a sufficient molecular insight to be transferred into pre-clinical studies. Please describe the commercially available Piezo1 agonists and their potential pre-clinical and clinical use. Currently, the Discussion section is limited to the molecular biology and KLF2/4 pathway regulation; add 2 or 3 paragraphs on the applications of your results. Which animal models should be used to test whether Piezo1 agonists are efficient in preventing endothelial dysfunction and atherosclerosis?

Author Response

On behalf of all the contributing authors, I sincerely thank the editor and the reviewers for their enthusiasm regarding our study and the insightful comments to improve the quality of this study. We performed additional experiments to address all the points raised by the reviewers. We have extensively revised our manuscript accordingly and marked revised manuscript with the changes. We hope that the changes we have made would resolve the reviewer’s concerns about the manuscript.

Here, Zheng and co-authors investigated the influence of mechanosensitive PIEZO1 channel on the regulation of atheroprotective transcription factors KLF2 and KLF4.  Overall, in vitro and in vivo experiments have been nicely performed and WBs are well complemented with qPCR data. The article is methodologically sound and is logically written. The authors employed an impressive armamentarium, including genetically engineered Cre-Lox mice with the endothelial-specific Piezo1 knockout and RNA sequencing of Yoda1-stimulated microvascular endothelial cells. All images and gels are of excellent quality.

The authors performed an important and timely discovery, convincingly showing the indispensable role of Piezo1 mechanosensitive channel in controlling the expression of KLF2 and KLF4, the major endothelial atheroprotective factors. This is particularly important when considering KLF2/4 agonists as promising tools to prevent the development of atherosclerosis at the gateway (i.e., to prevent endothelial dysfunction). PIEZO1 is a major mechanosensitive channel which can be finely targeted through the specific agonists (e.g., Yoda1 as was shown in the paper). The results can be therefore applicable in pre-clinical studies on ApoE or LDLR -/- mice.

My only significant concern is the statistics, as authors performed an incorrect parametric analysis (yet this does not deny the study results). The authors should revise the statistical analysis according to the respective comment (#2).

The paper can be accepted for publication upon the minor revision and is strongly recommended to be highlighted on the journal site or cover page.

Below are the specific comments:

  1. Rotational shaker is not the best option to induce shear stress as it is barely possible to calculate the exact shear stress values (dyn/cm2) in this experimental setting. A much better option is the cell culture under unidirectional flow perfectly imitating the blood flow (e.g. ibidi Pump System Quad), as it permits to set the exact shear stress values and even to simulate oscillatory and laminar flow. Yet, the authors ensured the disturbed or laminar flow by means of assessing the cell geometry.

Response: We appreciate the reviewer’s comments. The same issue was raised by other 2 reviewers. As reviewer’s suggestion, we have added the experiment using microfluidic system to mimics the fluid shear stress of bloodstream by controlling the flow rate in the revised Figure 1A. We found that the expression of Klf2 mRNA was rapidly induced at 3 hours after UF stimulation and followed by Klf4 induction at 6 hours with ~16 dyn/cm2 shear stress Figure 1A. This result was consistent with the data using rotating orbital shaker. We also added the experimental procedure in the revised Materials and Methods.

  1. The use of parametric statistics is incorrect for being used here, as the number of replicates is too low. Use non-parametric statistics instead (median, interquartile range, Kruskal-Wallis test, Mann-Whitney U-test). In any case, the use of standard error instead of standard deviation is also incorrect as it does not reflect the sample variation. Please also describe (and perform if needed) the adjustment for multiple comparisons (Tukey's post hoc test? False discovery rate?) Please also indicate all P values in a numerical manner, not as asterisks.

Response: Thank you for your consideration on the sample size and the testing method. The three biological replicates are more than enough to eliminate the random sampling bias. Meanwhile, the distribution of the data points is in mall interval, which highlights the significance of our results as well. Moreover, using t- test and three biological replicates are common statistical methods related with qPCR. General audience for this paper is the biochemist and cardiovascular expert rather than bio-statist. Therefore, using similar sample size and statistical approaches among publications made the comparison easier and more trustable.

As for the SD and SEM, we usually compare mean values, we are usually more interested in the estimates of the group means than in the group variance itself. Moreover, SD can be easily derived from SEM with the help of sample size when needed. To eliminate the error caused by in-consistent variance, the welch t-test was used. We have added detailed statistical analysis method in the revised manuscript and changed the asterisks with P value number.

  1. Please describe how you isolated RNA from mouse aortic endothelium to confirm the loss of PIEZO1 expression (Figure 2A).

Response: We apologize for the missing details. It has been shown KLF2/4 is highly expressed in endothelium of thoracic aorta [34], so we used whole aortic tissue to exact RNA to detect the expression of KLF2/4. To further conformed Piezo1 was deleted in endothelial cells, we isolated mouse brain endothelial cells from Tie2Cre;Piezo1f/f mice. We found that Piezo1 mRNA was significantly reduced in Piezo1-KO endothelium and KLF2/4 mRNAs were also markedly reduced in Piezo1-KO endothelium in the revised Figure 2C. We added the detail information how to isolate mouse thoracic aorta and brain endothelial cells in the revised Materials and Methods at page 4.

  1. Please comment on the clinical implications and translational potential of the study in the Discussion, as the study surely provides a sufficient molecular insight to be transferred into pre-clinical studies. Please describe the commercially available Piezo1 agonists and their potential pre-clinical and clinical use. Currently, the Discussion section is limited to the molecular biology and KLF2/4 pathway regulation; add 2 or 3 paragraphs on the applications of your results. Which animal models should be used to test whether Piezo1 agonists are efficient in preventing endothelial dysfunction and atherosclerosis?

Response: We appreciate for the reviewer’s suggestions on the potential clinical implication of our study. We have rewritten the discussion in the revised manuscript at page 16 and 17. 

Round 2

Reviewer 1 Report

no further comment